# The Effect of Silymarin Flavonolignans and Their Sulfated Conjugates on Platelet Aggregation and Blood Vessels Ex Vivo

**DOI:** 10.3390/nu11102286

**Published:** 2019-09-24

**Authors:** Jana Pourová, Lenka Applová, Kateřina Macáková, Marie Vopršalová, Thomas Migkos, Roger Bentanachs, David Biedermann, Lucie Petrásková, Václav Tvrdý, Marcel Hrubša, Jana Karlíčková, Vladimír Křen, Kateřina Valentová, Přemysl Mladěnka

**Affiliations:** 1Department of Pharmacology and Toxicology, Faculty of Pharmacy in Hradec Králové, Charles University, Heyrovského 1203, 500 05 Hradec Králové, Czech Republicapplovl@faf.cuni.cz (L.A.); voprsalova@faf.cuni.cz (M.V.); migkost@faf.cuni.cz (T.M.); bentanachs96@gmail.com (R.B.); tvrdyvac@faf.cuni.cz (V.T.); hrubsam@faf.cuni.cz (M.H.); 2Department of Pharmaceutical Botany, Faculty of Pharmacy in Hradec Králové, Charles University, Heyrovského 1203, 500 05 Hradec Králové, Czech Republic; macakovak@faf.cuni.cz (K.M.); karlickova@faf.cuni.cz (J.K.); 3Department of Pharmacology, Toxicology and Therapeutic Chemistry, School of Pharmacy and Food Science, University of Barcelona, Avda. Joan XXII 27-31, 08028 Barcelona, Spain; 4Institute of Microbiology of the Czech Academy of Sciences, Vídeňská 1083, 142 20 Prague, Czech Republic; biedermann@biomed.cas.cz (D.B.); petraskova@biomed.cas.cz (L.P.); kren@biomed.cas.cz (V.K.)

**Keywords:** milk thistle, *Silybum marianum*, sulfates, metabolites, vasorelaxant, aorta, thrombocytes, blood coagulation

## Abstract

Silymarin is a traditional drug and food supplement employed for numerous liver disorders. The available studies indicate that its activities may be broader, in particular due to claimed benefits in some cardiovascular diseases, but the contributions of individual silymarin components are unclear. Therefore, we tested silymarin flavonolignans as pure diastereomers as well as their sulfated metabolites for potential vasorelaxant and antiplatelet effects in isolated rat aorta and in human blood, respectively. Eleven compounds from a panel of 17 tested exhibited a vasorelaxant effect, with half maximal effective concentrations (EC_50_) ranging from 20 to 100 µM, and some substances retained certain activity even in the range of hundreds of nM. Stereomers A were generally more potent as vasorelaxants than stereomers B. Interestingly, the most active compound was a metabolite—silychristin-19-*O*-sulfate. Although initial experiments showed that silybin, 2,3-dehydrosilybin, and 2,3-dehydrosilychristin were able to substantially block platelet aggregation, their effects were rapidly abolished with decreasing concentration, and were negligible at concentrations ≤100 µM. In conclusion, metabolites of silymarin flavonolignans seem to have biologically relevant vasodilatory properties, but the effect of silymarin components on platelets is low or negligible.

## 1. Introduction

The dietary utilization of the milk thistle (*Silybum marianum* (L.) Gaertn., Asteraceae) was probably first mentioned by the ancient Greeks (Theophrastus of Eresos, ca. 371–287 BC and Pedanios Dioscurides, ca. 40–90 AD), with medicinal properties being described even in medieval times (Hildegard von Bingen). Liver protection has been the most popular application, encompassing traditional use against *Amanita phalloides* poisoning and snake bites, as well as the treatment of inflammatory liver diseases, liver cirrhosis, hepatitis C, and other liver pathologies. In fact, the milk thistle represents the most often recommended herbal product for patients with chronic liver diseases [1]. Silymarin, an extract of the milk thistle fruits (*cypselae*), has been largely tested for many other pharmacological effects, including those on metabolic syndrome [2,3,4]. This pathological condition is characterized by a complex of metabolic and other risk factors (hypertension, dyslipidemia, insulin resistance, diabetes, abdominal obesity, and pro-inflammatory and pro-thrombotic states), and represents a rising global health problem, which can progress to severe clinical complications such as acute myocardial infarction. Silymarin seems to target principal risk factors of metabolic syndrome and the mechanism of its action is likely complex. It involves the stabilization of cellular membranes, regulation of cell permeability, direct antioxidant activity, a reduction in insulin resistance and restoration of pancreatic beta cell function [4], antagonism at the human angiotensin (AT_1_) receptor [5], normalization of the function of the endothelium and vascular elasticity [6,7], and anti-atherosclerotic effects [8]. In studies with rats or mice, silymarin intake resulted in cardioprotective [9,10], antidiabetic, anti-inflammatory, and hypolipidemic effects [11,12,13,14]. Recently, silymarin was also reported to influence human blood coagulation by decreasing the platelet aggregation via inhibition of the cyclooxygenase (COX) activity [15]. Overall, silymarin seems to be generally beneficial to health: it is a popular component of various food supplements and has an excellent safety profile, as oral doses up to 2.1 g per day are tolerated well without adverse effects [16].

Chemically, silymarin is a mixture of natural substances with a slightly variable composition [17]. It contains approximately 70–80% flavonolignans (e.g., silybin (syn. silibinin), isosilybin, silychristin, isosilychristin, silydianin), and other flavonoids (e.g., taxifolin, quercetin, apigenin). The remaining 20–30% is represented by a relatively undefined polymeric flavonoid fraction. Flavonolignans (except silydianin) exist in silymarin as diastereomeric pairs referred to as A and B in various ratios [18]. There are limited data on the effects of (optically) pure silymarin components and their metabolites on the cardiovascular system.

The aim of this study was to investigate two types of potential cardiovascular effects of isolated silymarin flavonolignans and of their sulfated metabolites. Briefly, these compounds were screened ex vivo for vasorelaxant properties on isolated rat aorta, and for antiplatelet effects in human blood. Whenever possible, isolated diastereomers were used to address possible differences in their pharmacological action.

## 2. Materials and Methods

### 2.1. Animals

The experiments were carried out on male normotensive Wistar:Han rats purchased from Charles River (Düsseldorf, Germany). The animals were bred in the animal house of the Faculty of Pharmacy in Hradec Králové (Charles University) and kept at a temperature of 23–25 °C with a 12-h dark/light cycle. Rats were provided a standard diet and tap water ad libitum. The study (reg. No. MSMT-7041/2014-10) was approved by the Czech Ministry of Education, Youth and Sports and conformed to the Guide for the Care and Use of Laboratory Animals published by the US National Institutes of Health (8th edition, revised 2011, ISBN-13: 978-0-309-15400-0).

### 2.2. Silymarin Flavonolignans

All of the tested silymarin flavonolignans and their metabolites were isolated or synthetized at the Institute of Microbiology of the Czech Academy of Sciences in Prague (IM CAS), as reported below. Silybin was isolated from silymarin (Liaoning Senrong Pharmaceutical, Panjin, China, batch No. 120501) by its quick suspension in methanol and filtration, yielding solid silybin A+B (49.8% of silybin A, 48.0% of silybin B). Silybin diastereomers were chemoenzymatically resolved by immobilized lipase B from *Candida antarctica* (Novozyme 435, Novo-Nordisk, Copenhagen, Denmark) [19]. A series of consecutive acetylations and solvolyses was used to obtain silybin A (99.2%) and silybin B (99.0%). Silychristin (87.1% of silychristin A, 9.2% of silychristin B) was isolated from silymarin by Sephadex LH-20 chromatography as described in [20]. The 2,3-dehydro-derivatives, i.e., 2,3-dehydrosilybin (racemate, 98.2%), 2,3-dehydrosilybin A (95.1%), 2,3-dehydrosilybin B (97.4%), and 2,3-dehydrosilychristin (91.2%, containing 8.8% of silychristin A) were prepared by oxidation of the respective parent compounds as published previously [21,22]. Silybin A 20-*O*-sulfate (93%), silybin B 20-*O*-sulfate (99.9%), silychristin-19-*O*-sulfate (99.9%), 2,3-dehydrosilybin-20-*O*-sulfate (98%), 2,3-dehydrosilybin-7,20-*O*-disulfate (96%), 2,3-dehydrosilybin A 20-*O*-sulfate (98%), and 2,3-dehydrosilybin B 20-*O*-sulfate (94%) were prepared using aryl sulfotransferase from *Desulfitobacterium hafniense* according to the previously described procedure [22]. The structures of all compounds tested are shown in Figure 1, NMR and MS spectra of all compounds used were identical to authentic standards available in the Laboratory of Biotransformation, IM CAS; HPLC chromatograms are presented in Appendix A.

### 2.3. Reagents

Urethane, norepinephrine bitartrate, phenylephrine, sodium nitroprusside, acetylcholine, dimethyl sulfoxide (DMSO), indomethacin, ethylenediaminetetraacetic acid (EDTA), terutroban, acetylsalicylic acid, 1-benzylimidazole, and kaempferol were purchased from Sigma-Aldrich (Prague, Czech Republic). The Krebs solution salts and 96% ethanol were purchased from Penta s.r.o. (Prague, Czech Republic). Arachidonic acid was bought from Medista (Prague, Czech Republic). Heparin sodium was purchased from Zentiva (Prague, Czech Republic); U-46619, thromboxane B_2_ ELISA kit, prostaglandin H_2_, and the COX inhibitor screening kit were from the Cayman Chemical Company (Ann Arbor, MI, USA). Collagen was obtained from Diagnostica a.s. (Prague, Czech Republic), and saline from B. Braun (Prague, Czech Republic).

### 2.4. Ex Vivo Experiments on Isolated Rat Aortas

For these experiments, the rats were anesthetized with urethane (1.2 g·kg^−1^ injected intraperitoneally), and exsanguinated via the abdominal aorta. The thoracic aorta was very gently excised from the rat, and transferred to a Petri dish. It was cleaned of connective tissue and blood, and then cut into rings measuring 3 mm in length. When needed for mechanistic studies, the endothelial layer was mechanically disrupted by gently rubbing the luminal surface with dental floss. The rings were maintained in tissue baths with Krebs solution (in mM: NaCl 94.8, KCl 4.7, KH_2_PO_4_ 1.18, MgSO_4_·7H_2_O 1.18, CaCl_2_·2.5, NaHCO_3_ 25, D-glucose 11.6) oxygenated with carbogen (95% O_2_/5% CO_2_) at 37 °C. Each aortic ring was placed between two stainless-steel wire hooks, one of them rigidly attached to the end of a fixed support rod and the second connected to a transducer and a computer equipped with S.P.E.L. Advanced kymograph Software (Experimetria Ltd., Budapest, Hungary). This arrangement enabled the measurement of tissue contraction or relaxation.

Aortic rings were stretched at a tension of 2 g and equilibrated for 40 min and washed with Krebs solution every 10 min. Thereafter, the tissue baths were filled with another 5 mL of Krebs solution. The aortic rings were contracted with norepinephrine (the final concentration in the bath was 1 µM). To confirm an intact endothelium, acetylcholine (10 µM) was added to the bath when the contraction became stable. The endothelium-free vessels did not respond to acetylcholine. After that, acetylcholine and norepinephrine were completely removed by repeated washing with Krebs solution.

In the next step, each tissue bath was filled with another 5 mL of Krebs solution and the aortic rings were contracted with phenylephrine (1 µM). After stabilization, the tested compound was cumulatively added to the bath to final concentrations ranging from 100 nM to 1 mM. The next dose was always applied after stabilization of the previous response. At the end of each experiment, maximal relaxation was induced by adding sodium nitroprusside (10 µM) to the bath. This concentration produces 99% vasorelaxation of rat aorta, and was calculated using the vasorelaxant curve of sodium nitroprusside obtained in our pre-experiments.

The tested substances were dissolved in DMSO, and the final concentration of DMSO in the bath never exceeded 2%. As the negative control, the same concentrations of DMSO but without the tested substance were cumulatively added to several aortic rings in each experiment.

### 2.5. Platelet Aggregation Experiments

#### 2.5.1. Blood Volunteers

Blood samples from 22 healthy, non-smoking volunteers were collected by venipuncture into plastic disposable syringes containing heparin sodium (170 IU/10 mL). Whole blood was used for all measurements in an impedance aggregometer Multiplate (Roche Diagnostic, Basel, Switzerland). For other experiments, platelet-rich plasma was obtained as the supernatant after centrifuging the blood for 8 min at 214× *g* (VWR Compact Star CS4 centrifuge, VWR International Ltd., Lutterworth, U.K.); the number of platelets was counted in a Neubauer Improved counting chamber (Marienfeld, Lauda-Königshofen, Germany) with the use of an inverted Nikon Eclipse TS100 microscope (Nikon Corporation, Tokyo, Japan) and adjusted to 3.5 × 10^8^ per mL by autologous platelet-poor plasma prepared by further centrifugation of the remaining blood at 2771× *g* for 10 min. The COX inhibitor indomethacin (at a final concentration of 10 µM), or the thromboxane synthase inhibitor 1-benzylimidazole (20 µM) were added immediately after the blood collection for experiments focusing on the inhibition of thromboxane synthase and COX, respectively. The study was approved by the Ethics Committee of Charles University, Faculty of Pharmacy in Hradec Králové (approval date: November 12, 2012) and conforms to the latest Helsinki Declaration. All volunteers signed written informed consent for the study.

#### 2.5.2. Platelet Aggregation Induced by Collagen and Arachidonic Acid

The aggregometer Multiplate was used for the measurements. Briefly, 300 µL of whole blood were diluted with the same volume of preheated 0.9% sodium chloride and incubated with 5 µL of the tested compound dissolved in DMSO (at a final concentration of 0.8%) for 3 min at 37 °C. Platelet aggregation was then induced with collagen or arachidonic acid and monitored for 6 min. The dose of the inducer was first set to the minimal concentration, which caused maximal aggregation, and the second calibration was carried out using acetylsalicylic acid (ASA) and kaempferol to fit our standard curves. The final concentrations of collagen and arachidonic acid were in the ranges of 0.41–1.22 μg·mL^−1^ and 86–196 μM, respectively.

#### 2.5.3. Cyclooxygenase-1 (COX-1) Inhibition

COX-1 inhibition was determined with ELISA using a commercial kit from Cayman Chemicals. ASA or the tested compounds were incubated with ovine COX-1 at 37 °C and then arachidonic acid (at a final concentration of 100 µM) was added to start the reaction. After 2 min, the reaction was stopped with hydrochloric acid and the amount of prostaglandin H_2_ formed was measured after its reduction to prostaglandin F_2α_ by SnCl_2_, according to the COX inhibitor screening kit manual [23]. The percentage inhibition was relative to the positive control, which contained only the solvent and arachidonic acid.

Analogously, platelet-rich plasma pretreated with 1-benzylimidazole (see above) to block further metabolism of prostaglandin H_2_ was used to test the inhibition of human COX-1 instead of ovine COX-1. ASA was also used as a standard here. The percentage of inhibition was calculated as in the previous test.

#### 2.5.4. Thromboxane A_2_ Synthase Inhibition

Thromboxane A_2_ synthase inhibition was evaluated according to the method of Chang et al. [24] with minor modifications. Indomethacin-containing platelet-rich plasma was incubated with the tested compound for 3 min at 37 °C. After the addition of prostaglandin H_2_ (50 ng), the mixture was incubated precisely for 5 min. The reaction was terminated by the addition of cold EDTA solution (2 mM, 4 °C) and the mixture was centrifuged at 10,500 *g* for 2 min. The thromboxane B_2_ levels in the supernatants were measured using a commercial kit [25]. 1-Benzylimidazole was used as a standard. The percentage inhibition was relative to the positive control, which contained only the solvent and prostaglandin H_2_.

#### 2.5.5. Antagonism at Thromboxane A_2_ Receptors

Antagonism at thromboxane A_2_ receptors was analyzed in the aggregometer Multiplate. The inducer of aggregation was U-46619, a stable agonist of thromboxane A_2_ receptors. The final concentration of U-46619 was 1.09 μM; terutroban was used as a standard.

### 2.6. Statistical Analysis

GraphPad Prism 7.03 (GraphPad Software, San Diego, CA, USA) was used for all data analysis. Results were analyzed by comparing 95% confidence intervals of vasorelaxant curves and by EC_50_ followed by ANOVA with the Tukey multiple comparison test.

## 3. Results

### 3.1. Ex Vivo Experiments on Isolated Rat Aorta

The majority of the 15 tested flavonolignans, including optically pure compounds, racemates and their metabolites, exhibited a vasorelaxant effect on rat aorta; however none of them induced a complete vasorelaxation at the tested concentrations ranging from 100 nM to 1 mM. Silychristin-19-*O*-sulfate, silybin A, silychristin, 2,3-dehydrosilybin A, and silybin exhibited the highest vasorelaxant activities, with detectable vasorelaxant effects as low as hundreds of nM (see Figure 2) and half maximal effective concentrations (EC_50_ values) between 19 and 30 μM. In contrast, 2,3-dehydrosilybin B, silybin A 20-*O*-sulfate and 2,3-dehydrosilybin-7,20-*O*-disulfate were not effective (Figure 3). The activity of stereoisomers A with the 10*R*,11*S*-configuration was in general higher than that of B stereoisomers (10*S*,11*S*; Appendix A). The activity of their equimolar mixture was in between that of individual stereoisomers (Appendix A). We also evaluated the effect of sulfation on the vasorelaxant activity, and the results show that monosulfation in contrast to disulfation did not decrease the activity (Appendix A).

The most potent vasorelaxant non-conjugated flavonolignan silybin A was chosen for subsequent analysis of the role of the endothelium in its activity. The data from endothelium-denuded rat aortic rings clearly show that the removal of the endothelium from the rings abolished its vasorelaxant activity (Figure 4).

### 3.2. Platelet Aggregation Experiments

The first screening of antiplatelet activity was performed using collagen as the trigger and the standard antiplatelet drug acetylsalicylic acid (ASA) as a benchmark. The initial experiments were performed with high final concentrations (120 and 240 μM) to select the most active compounds (Figure 5). Six substances reached the same activity as ASA at 240 μM (Figure 5), and 2,3-dehydrosilychristin was even more active than ASA. Nevertheless, only silybin A had an activity comparable to ASA at 120 μM, and all compounds exhibited a steep loss of activity at lower concentrations (Appendix A).

The same setup was used to measure the effect of all flavonolignans on platelet aggregation induced by arachidonic acid, a downstream player in the collagen aggregation pathway. Only silybin B, silybin B-20-*O*-sulfate and 2,3-dehydrosilybin exhibited an effect similar to that of ASA at 120 and 240 μM (Figure 6). Similarly to experiments using collagen as the trigger, the loss of activity of all three substances was very steep, resulting in almost no activity at 40 μM (vs. 65% inhibition by ASA, Appendix A). The stereoisomers B were significantly more active than the stereoisomers A (Appendix A).

The most active substances from the initial screening were subjected to a series of mechanistic tests evaluating their effects on the arachidonic acid platelet aggregation cascade. The transformation of arachidonic acid to prostaglandin H_2_ by COX-1 is the initial step in this cascade and therefore the substances were first tested for their inhibition of this enzyme using recombinant ovine COX-1. The only substance that inhibited the enzyme in a similar manner to ASA was 2,3-dehydrosilybin (Figure 7b) with IC_50_ 25.2 ± 10.4 µM (*cf*. 79.2 ± 27.2 μM for ASA). Other compounds were significantly less active (Figure 7a). Interestingly, the racemate was more potent than pure stereoisomers (Figure 7b). To confirm the effect of 2,3-dehydrosilybin in a more clinical setting, human platelets were used as a source of COX. The effect of this flavonolignan was however low and ASA was a stronger inhibitor in this model compared with ovine COX-1 (data not shown).

The second step in the arachidonic acid cascade is the transformation of prostaglandin H_2_ to thromboxane A_2_ (TXA_2_) by thromboxane A_2_ synthase, and so the active substances were also tested for their inhibition of this enzyme at 100 μM compared to 1-benzylimidazole. 2,3-Dehydrosilychristin was the only substance to exhibit an inhibitory activity towards this enzyme (Appendix A). Since this compound was not very active in terms of arachidonic acid-induced aggregation, this effect at such a high concentration was not considered to be important. The range of activities of other substances were within the error of the method and hence insignificant.

Since silybin B, silybin-B-20-*O*-sulfate, and 2,3-dehydrosilybin only exhibited a weak inhibition of COX-1 and no inhibition of thromboxane A_2_ synthase, we also measured their inhibition of platelet aggregation induced by U46619, an agonist of thromboxane receptors. Silybin B and 2,3-dehydrosilybin exhibited dose-dependent antagonistic effects, while silybin B-20-*O*-sulfate was almost inactive (Appendix A).

## 4. Discussion

Silymarin has a long history of therapeutic use, with hepatoprotection being the most important of its applications. Besides other effects, recent reviews summarized the promising potential of silymarin for the prevention and treatment of metabolic syndrome [2,4]. The spectrum of positive effects of silymarin towards metabolic syndrome is very broad and encompasses lipid lowering, anti-atherosclerotic, antidiabetic, and other positive cardiovascular effects. The latter might be at least partly based on vasorelaxant and antiplatelet activities, the main research objectives of this work.

Pure substances are pivotal for determining the contribution of individual silymarin flavonolignans to the observed effect. As silymarin flavonolignans are extensively metabolized, it was also important to test selected human metabolites as well. Phase I has been repeatedly shown to play a marginal role in silymarin metabolism [26,27]. Crucial reactions are those of phase II, and the formed conjugates (glucuronides, sulfates) represent a significant part of the total plasma level of silymarin constituents. Only 12% of silybin A and 3% of silybin B remained unconjugated after their intragastric administration to rats [27,28,29,30]. Therefore, six sulfates and one disulfate were prepared [22] and tested in this work to compare their activity with those of the respective parent compounds.

Stereoselectivity is apparently an important factor in biological systems. For silymarin, individual diastereomers were reported to be subject to different metabolism in humans. In particular, the same doses of silybin A and silybin B reached different plasma levels, and were metabolized to a different extent, with different velocity, and yielded non-identical spectra of conjugates in rats [31]. Analogously, isosilybin diastereomers also exhibited different metabolisms [28,29,31,32]. The same may be true for their pharmacodynamic effects. Hence in this study, both equimolar mixtures (racemates) and pure stereoisomers A and B were prepared and tested. Silychristin was an exception, as natural silychristin containing 89% diastereomer A and 9% B was used, which could be considered to be silychristin A of nearly 90% purity.

### 4.1. Ex Vivo Experiments on Isolated Rat Aorta

The vasorelaxant properties of silymarin flavonolignans were addressed ex vivo on isolated rat aorta. This is a classical experimental model widely used in basic pharmacological research [30,33,34]. The parent flavonolignans were studied first. Silybin A, silychristin and 2,3-dehydrosilybin A exhibited notable vasorelaxant effects, while 2,3-dehydrosilybin B was devoid of that activity (Figure 3). Although the effects observed are incomparable with clinically used medication (EC_50_ value of silybin A was 26.8 μM; that of a calcium channel blocker nifedipine is 4.47 nM [35]), these results deserve attention since we are considering effects of a natural product used as a food-supplement, and not an approved drug. In particular, the effect of silybin A is noteworthy, as silybin is clearly the predominant component of the silymarin complex (approximately 30%), and it is also believed to be the main active substance of silymarin—which is nevertheless questionable [17,36]. Importantly, the concentrations at which the most active compounds reached a 50% vasorelaxant effect (EC_50_) were in the tens of µM, and this is achievable in plasma using novel forms of administration [26,37]. Moreover, slight but observable vasodilatory effects were present even at units of µM, and for silychristin at hundreds of nM (Figure 2). Indeed, there are some animal studies that show silymarin or silybin to decrease blood pressure: Antihypertensive effects of silymarin have been previously found in unilateral nephrectomized rats with deoxycorticosterone acetate (DOCA)-induced hypertension (300 or 500 mg·kg^−1^ of silymarin orally for 4 weeks) [38], and those of silybin (300 mg daily orally for 8-12 days) in spontaneously hypertensive rats subjected to acute coronary artery occlusion [39]. Positive effects of silymarin supplementation were also reported in humans, namely in patients with hypertension and microalbuminuria. The adjuvant use of silymarin (420 mg daily orally for 2 months) along with antihypertensive drugs (atenolol and furosemide) led to a drop in blood pressure, an improvement in lipid profile, and reduced microalbuminuria vs. placebo [40]. These results could be attributed to the regulation of vascular tonus. Moreover, silybin administered to obese diabetic mice (20 mg·kg^−1^ i.p. daily for four weeks) was found to improve endothelial dysfunction, a well-known crucial factor for the maintenance of healthy blood vessels [7].

In our experiments, the vasorelaxant effects of silymarin flavonolignans were clearly dependent on the stereomeric configuration of the compounds under study. The stereomers B were generally less potent than the stereomers A (silybin), or even ineffective (2,3-dehydrosilybin, Appendix A). To our knowledge, no other study has addressed the stereoselectivity of silymarin flavonolignans in relation to either vasorelaxant activity or cardiovascular effects. Additionally, in this work, some substances were tested as equimolar mixtures to verify possible interactions between both diastereomers. No potentiation or inhibition of vasorelaxant potency was observed. In all cases, the EC_50_ values of the mixture were between the EC_50_ values of the individual diastereomers (Appendix A). However, the interactions among flavonolignans certainly deserve further investigation, as silymarin is a complex mixture that contains various isomeric compounds including diastereomers.

The effects of sulfated conjugates were often comparable to those of the parent compounds (e.g., 2,3-dehydrosilybin A-20-*O*-sulfate) or were slightly more potent (2,3-dehydrosilybin-20-*O*-sulfate and silychristin-19-*O*-sulfate). Silychristin sulfate was even the most efficient substance tested. However, the differences between active parent flavonolignans and their sulfates were generally negligible. On the other hand, the presence of two sulfate groups (2,3-dehydrosilybin-7,20-*O*-disulfate) fully abolished the vasorelaxant effect of 2,3-dehydrosilybin (Appendix A). The clinical importance of this finding is uncertain, since the combined conjugates (including disulfates) are produced as minor metabolites [41]. Vasorelaxation can occur by various mechanisms, both endothelium-dependent and endothelium-independent. We have shown that the vasorelaxation induced by silybin A was clearly dependent on the presence of intact endothelium. Further experiments are necessary to elucidate this effect in details (e.g., NO generation, role of M receptors and/or of various potassium or L-type calcium channels). Less expected mechanisms such as inhibition of phosphodiesterases previously reported for some natural flavonoids [34] are also possible.

### 4.2. Platelet Aggregation Experiments

Platelet aggregation is a very complex process, essential for maintaining vascular hemostasis. The normal endothelium provides a non-adhesive surface for platelets. However, after vascular injury, platelets are exposed to subendothelial collagen fibers, which are responsible for the initial platelet adhesion phase [42]. Hence in our initial screening, we employed collagen-triggered platelet aggregation in the whole blood as the most suitable physiological model. Some previous results reporting antiplatelet activity are available for a few silymarin components. Silybin and silychristin were shown to inhibit collagen-induced platelet aggregation in platelet-rich plasma and blood platelet aggregate formation in whole blood, and to decrease the expression of activation markers on their surface, such as P-selectin and an active form of αIIbβ3. Silychristin was more potent than silybin in all tests, and both substances inhibited platelets’ adhesion to collagen fibers as well [43]. In our study, silybin exhibited a slightly stronger antiplatelet activity than silychristin. However, the difference between them was not significant, and both substances were obviously less active than acetylsalicylic acid (ASA), used as a standard antiplatelet drug (Figure 5). The discrepancy between these two studies could lie in the difference between platelet-rich plasma [43] and the whole blood employed in our study. In addition, no clinically used drug was used in the above study, so the comparison is not straightforward. Interesting results were obtained with the individual diastereomers of silybin. Both substances at 240 μM inhibited collagen-induced aggregation in a manner comparable to that of ASA, but the loss of effect with decreasing concentration was much steeper (Appendix A), exhibiting no activity at achievable plasma levels [41]. The activities of diastereomers were comparable. 2,3-Dehydrosilychristin, the most active substance, was a significantly stronger inhibitor than silychristin, indicating the importance of the 2,3-double bond in the flavonolignan skeleton for collagen-induced aggregation. However, the same phenomenon was not observed with 2,3-dehydrosilybin and silybin, suggesting a more complex structure-activity relationship.

Importantly, sulfation did not significantly decrease the effect with the exception of silybin A and 2,3-dehydrosilybin, whose conjugation resulted in a significant drop in activity. All other sulfates and 2,3-dehydrosilybin-7,20-*O*-disulfate inhibited collagen-induced platelet aggregation to the same extent as their parent compounds (Appendix A).

The collagen cascade is tightly connected with the arachidonic acid cascade, but also triggers other pathways. Silymarin and its isolated flavonolignans previously decreased the platelet aggregation initiated by other inducers such as arachidonic acid or adenosine diphosphate (ADP) [15,44,45,46]. Silychristin and silybin strongly reduced arachidonic acid-induced aggregation in platelet-rich plasma at the highest concentration tested (100 μM) to 13.3% and 24% of the control, respectively. Furthermore, silychristin exhibited a poor but significant activity even at plasma-achievable 10 μM [15]. These findings are not consistent with our whole blood tests, where silychristin did not exhibit any significant antiaggregatory activity compared to the control (DMSO) nor at the highest concentration tested (240 μM), and silybin exhibited weak inhibitory activity at the highest concentration (Figure 6). The accepted fact is that not only platelets, but also red blood cells play a crucial role in blood clotting [47]; hence the confirmation of the results in whole blood is physiologically more relevant.

The activity of silybin B clearly exceeded the effect of silybin A, suggesting that its stereochemistry is crucial for antiaggregatory activity at this level. This was previously also shown for other phenolic substances with isomeric structures: catechin and epicatechin [48], and was demonstrated for the vasorelaxant effect in this study. However, in terms of vasorelaxant effect, these two diastereomers showed the opposite trend (silybin B was significantly less active than silybin A); the activity of racemic silybin was between that of its pure diastereomers (Figure 6 and Appendix A). The same tendency is also visible for other pairs of stereoisomers tested in this study (Figure 6). In contrast, the presence or absence of the 2,3-double bond does not play any role in arachidonic acid-induced platelet aggregation, as racemic silybin and 2,3-dehydrosilybin did not exhibit a significantly different effect. Similarly, no difference was found between individual diastereomers and their 2,3-dehydroderivatives in their effects on arachidonic acid-induced aggregation.

Most sulfated conjugates were as active as their parent compounds, which is a fundamental finding for the most active substance silybin B and its sulfate, because silybin B is conjugated faster and to a significantly greater extent than silybin A [31]. The only negative effect of sulfation was noted for both 2,3-dehydrosilybin A and B, where it led to an almost complete loss of activity (Figure 6).

To reveal the mechanism of action of the most active substances, they were subjected to a series of tests evaluating their effect on individual steps of the arachidonic acid pathway. Arachidonic acid released by phospholipase A_2_ from the platelet membrane is rapidly transformed to prostaglandin H_2_ by COX-1. ASA, used in antiplatelet therapy, inhibits COX-1 activity while being inactive in further steps of this pathway. Therefore, silymarin flavonolignans were tested for COX-1 inhibition. Silybin and silychristin previously substantially inhibited human platelet COX-1 both in vitro and in silico [15]. In contrast, both silybin diastereomers as well as their sulfates exhibited only very weak inhibitory activity towards ovine COX-1 in our study. The most potent ovine COX-1 inhibitor (even more than ASA) was racemic 2,3-dehydrosilybin. However, when human platelets were used as a source of COX-1 enzyme, the effect of 2,3-dehydrosilybin was markedly decreased in contrast to ASA, suggesting that COX-1 inhibition is clinically unimportant in this case. Such a difference in activity between the recombinant ovine and human platelet enzyme is not uncommon [48,49] and should always be tested. The highest effect was observed with racemic mixtures, suggesting a potentiation of the activity of individual stereoisomers. Such an effect seems to be rare, but a similar phenomenon was observed in a pharmacokinetic study. Silybin A metabolism with bovine liver microsomes was slower when incubated separately than when incubated together with silybin B [32].

A further step in the arachidonic acid cascade is the transformation of prostaglandin H_2_ to TXA_2_ by thromboxane synthase. Inhibition of this step was also suggested previously as a possible mode of action for silybin and silychristin [15]. Similar results for silybin were also obtained in another study where calcium ionophore A23187 was used as stimulant of TXA_2_ formation. The IC_50_ value was 69 μM [45]. Apparently, this observed result was not due to direct inhibition of thromboxane synthase, since the enzyme was not inhibited in our experiments, where we started the reaction by the addition of the physiological substrate, prostaglandin H_2_. Apart from 2,3-dehydrosilychristin, no substance was able to block the activity of the enzyme even at 100 μM. Based on the different results in arachidonic acid and collagen-triggered platelet aggregation, inhibition of this enzyme had likely only little relevance to our results.

When TXA_2_ is formed, it binds to thromboxane receptors as the next step in the arachidonic acid pathway. To the best of our knowledge, the antagonism of flavonolignans from silymarin on thromboxane receptors has not been studied yet. Only three of the most active substances were tested for their effect on U-46619 (an agonist of the above-mentioned receptor)-induced platelet aggregation. Silybin B and 2,3-dehydrosilybin B exhibited significant concentration-dependent inhibition of platelet aggregation at concentrations corresponding to those that were effective against arachidonic acid-induced platelet aggregation (Appendix A). Therefore, it is evident that this is the main mechanism of action of these two substances.

The results presented here may contradict some previous studies suggesting silymarin flavonolignans to be capable of significant platelet aggregation inhibition even at plasma-achievable concentrations [15,43,44,45,46]. The main weakness of the studies published earlier is the absence of a standard drug, which makes their comparison with our results difficult.

## 5. Conclusions

The cardiovascular effects of silymarin seem to be associated with its vasorelaxant activity rather than with its antiplatelet effects. Silybin A, silychristin (90% isomer A), and 2,3-dehydrosilybin A exhibited vasorelaxant effects ex vivo on isolated rat aorta at concentrations achievable in plasma. The vasorelaxant activity of monosulfates was similar to those of the corresponding parent substances, and the most active tested compound was a metabolite, silychristin-19-*O*-sulfate. The tested disulfate was inactive. Vasorelaxant effects were stereoselective, and A diastereomers were generally more potent than the B diastereomers. No potentiation or inhibition of activity was found when equimolar mixtures of diastereomers were tested. The vasorelaxation induced by silybin A was dependent on the presence of an intact endothelium; however, further experiments are necessary to elucidate this mechanism in detail.

In contrast to vasorelaxant effects, silybin B, its sulfate and 2,3-dehydrosilybin B were the only flavonolignans to exhibit antiplatelet activities, but these effects should be considered weak. The antagonism of silybin B and 2,3-dehydrosilybin B at thromboxane receptors is the main mechanism of action of these two substances. Despite these at first sight encouraging results, it is highly improbable that this activity would be manifested in vivo due to the high concentrations needed to evoke this effect.

## Figures and Tables

**Figure 1 nutrients-11-02286-f001:**
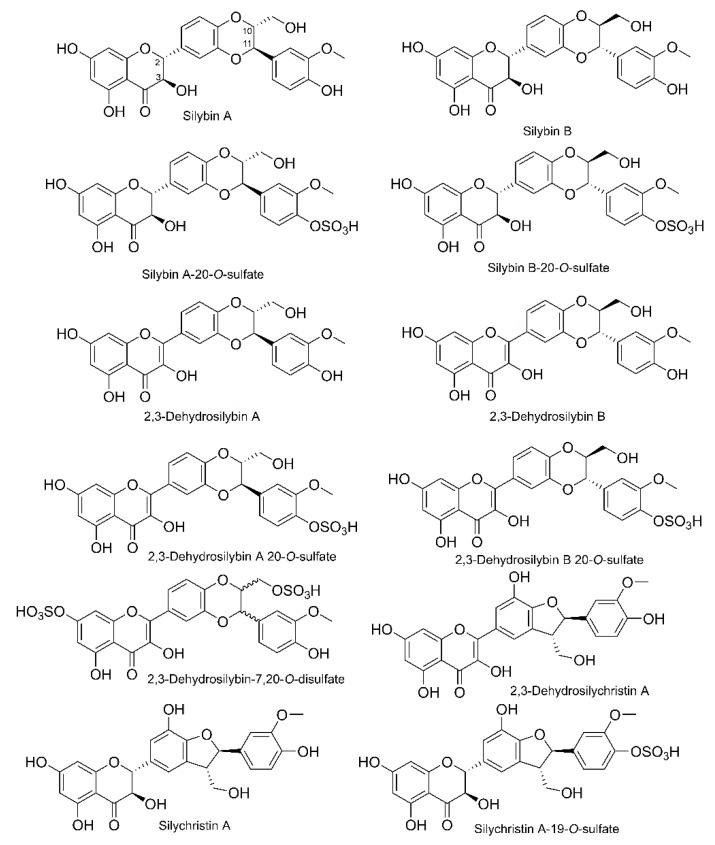
Structures of the compounds tested.

**Figure 2 nutrients-11-02286-f002:**
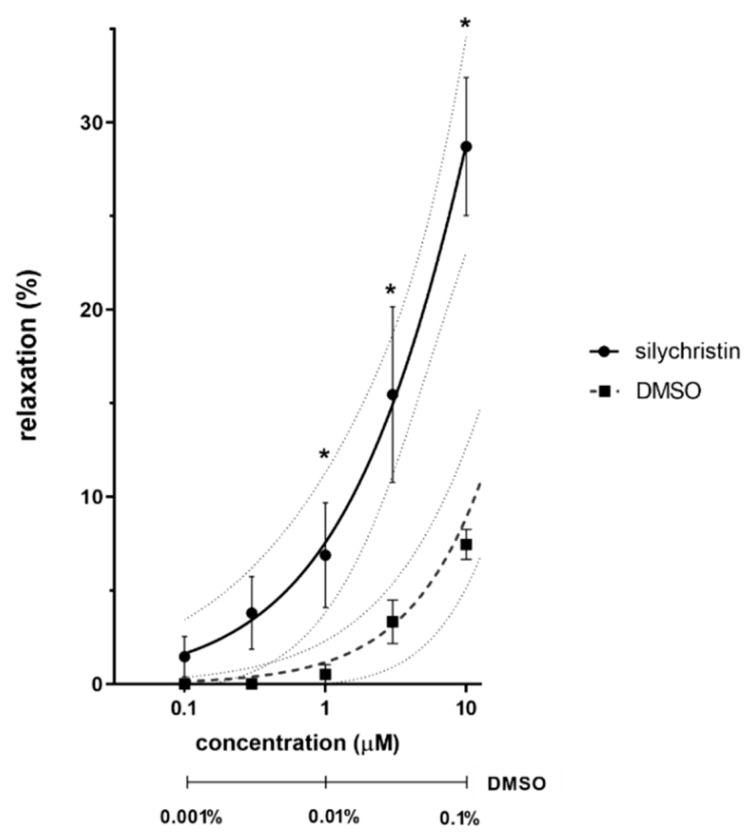
Vasodilatory effect of silychristin on aortic rings precontracted with phenylephrine. The percentage of relaxation was calculated using the standard vasorelaxant drug sodium nitroprusside, which produces 99% vasorelaxation at 10 µM. Data are expressed as means ± SEM, *n* = 5. The concentration of DMSO is shown below the x axis. * *p* < 0.05 vs. DMSO.

**Figure 3 nutrients-11-02286-f003:**
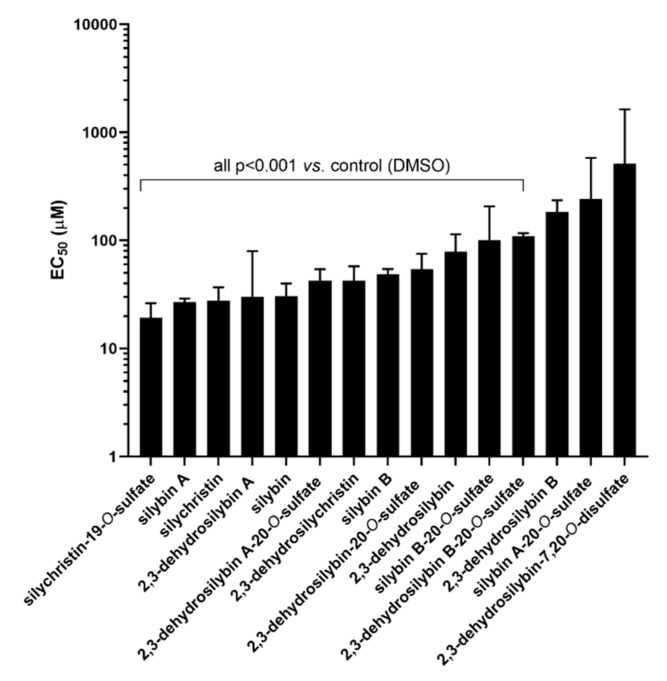
Ex vivo vasorelaxant effects of silymarin flavonolignans on intact rat aortic rings precontracted with phenylephrine. Data are expressed as EC_50_ values, with the error reflecting the 95% confidence interval; *n* = 4 with the exception of silychristin and 2,3-dehydrosilybin A-20-*O*-sulfate (*n* = 5); 2,3-dehydrosilybin A (*n* = 6); and 2,3-dehydrosilybin B-20-*O*-sulfate and 2,3-dehydrosilybin B (*n* = 3).

**Figure 4 nutrients-11-02286-f004:**
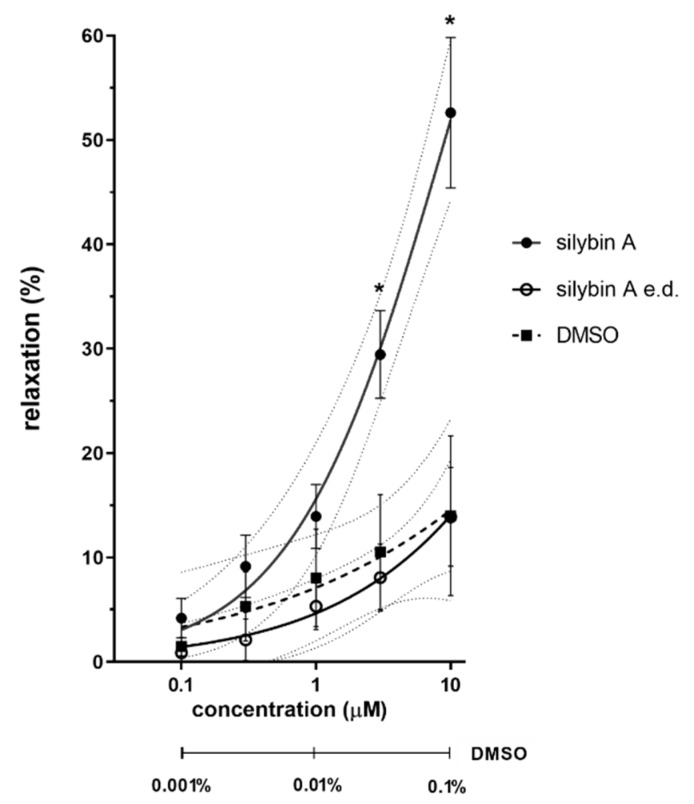
Role of endothelium in vasorelaxant activity of silybin A. The effect was studied on intact or endothelium-denuded (e.d.) rat aortic rings precontracted with phenylephrine. Data are expressed as means ± SEM, *n* = 4. The concentration of DMSO is shown below the x axis. * *p* < 0.05.

**Figure 5 nutrients-11-02286-f005:**
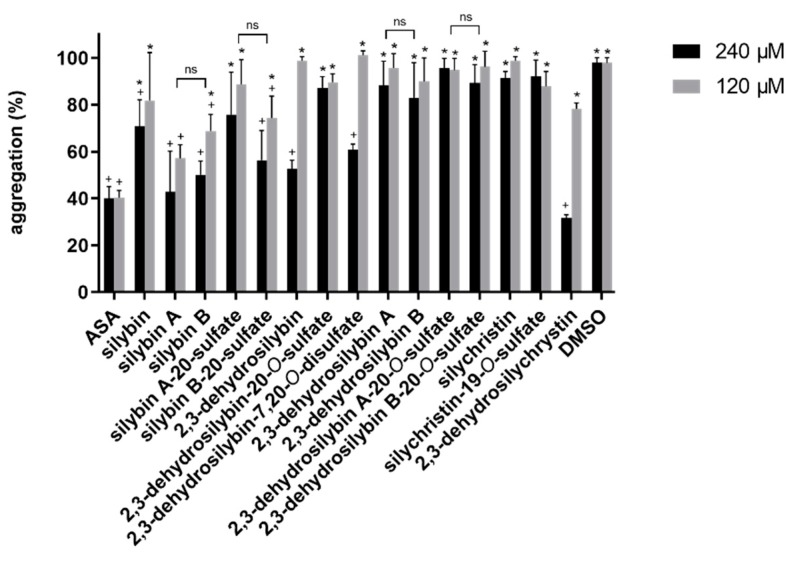
Effect of flavonolignans on whole-blood platelet aggregation induced by collagen. Data are expressed as mean ± SD. * *p* < 0.05 vs. acetylsalicylic acid (ASA); + *p* < 0.05 vs. solvent (DMSO); ns: non-significant.

**Figure 6 nutrients-11-02286-f006:**
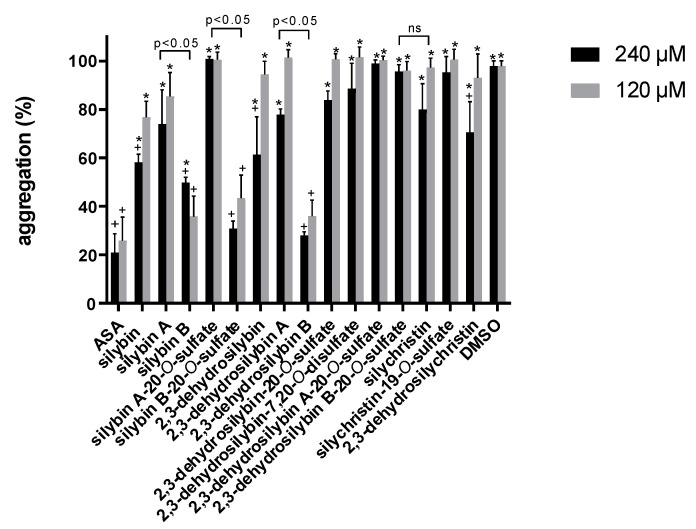
Effect of flavonolignans on whole-blood platelet aggregation induced by arachidonic acid. Data are expressed as mean ± SD. * *p* < 0.05 vs. acetylsalicylic acid (ASA); ^+^
*p* < 0.05 vs. solvent (DMSO); ns: *p* ≥ 0.05.

**Figure 7 nutrients-11-02286-f007:**
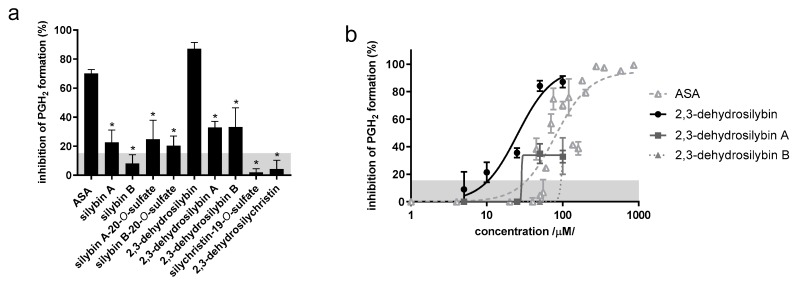
Effect of selected flavonolignans on ovine COX-1. (**a**): Comparison of tested substances and acetylsalicylic acid (ASA) at the final concentration of 100 μM, (**b**): Concentration-dependent curves of selected substances. * *p* < 0.01 vs. ASA. Grey area indicates the error of the method. Data are expressed as mean ± SD.

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
