# Peer review of "The Effect of Silymarin Flavonolignans and Their Sulfated Conjugates on Platelet Aggregation and Blood Vessels Ex Vivo"

_nutrients, 2019, doi:10.3390/nu11102286_

Round 1
Reviewer 1 Report
The manuscript by Pourová J et al., describes that The effect of silymarin flavonolignans and their sulfated conjugates on platelet aggregation and blood vessels ex vivo. This is an interesting possibility and likely to be potentially important information to prevent cardiovascular disease by functional food. For the benefit of the reader, however, some points need clarifying. My comments to this article were as follows.
The authors concluded that silymarin flavonolignans are associated with vasorelaxant activity rather than antiplatelet effects in this manuscript. How do silymarin flavonolignans exert vasorelaxant effects? Do silymarin flavonolignans affect receptor of relaxant agonist, nitric oxide synthase (NOS), and NO production in endothelium? In Figure 4 and 5, the authors used acethlsalicylic acid (ASA) as positive control. Similarly, the authors should show the positive control to compare effects of silymarin flavonolignans in Figure 2.
Author Response
How do silymarin flavonolignans exert vasorelaxant effect? Do silymarin flavonolignans affect receptor of relaxant agonist, nitric oxide synthase (NOS), and NO production in endothelium?
Response:
This is a very useful comment and we agree that the mechanism of action represents a crucial pharmacological aspect. We have shown that the vasorelaxation induced by silybin A is clearly dependent on the presence of intact endothelium. We plan to elucidate the mechanism of this vasorelaxation in details (e.g. dependence on NO generation, M receptor activation, role of potassium and/or calcium channels using standard antagonists such as L-NAME, atropine, TRAM-34, UCL-1684 or activators as BayK8644). These experiments however require a substantial time and animals, so waiting for finishing of this analysis would significantly delay the publication (we are supposed to revise this manuscript within 5 days). For this reason, we plan to perform this complex analysis during our future studies.
The following text was added to the Discussion (at L. 347):
Vasorelaxation can occur by various mechanisms, both endothelium dependent and endothelium independent. We have shown that the vasorelaxation induced by silybin A was clearly dependent on the presence of intact endothelium. Further experiments are necessary to elucidate this effect in details (e.g. NO generation, role of M receptors and/or of various potassium or L-type calcium channels). Less expected mechanisms such as inhibition of phosphodiesterases previously reported for some natural flavonoids are also possible.
In Figure 4 and 5, the authors used acetylsalicylic acid (ASA) as positive control. Similarly, the authors should show the positive control to compare effects of silymarin flavonolignans in Figure 2.
Response:
We agree that inclusion of a positive control in figures can improve the comprehensibility and comparability of results. In our ex vivo experiments, maximal relaxation of rat aorta was induced by adding sodium nitroprusside (10 μM) to the bath. This concentration produces 99% vasorelaxation of isolated rat aorta, and was derived from our previous experiments (vasorelaxant curve of sodium nitroprusside with calculation of EC50 and EC99). We prefer to present our results in Figure 2 without this positive control, as the difference between EC50 values is enormous. It is not surprising, sodium nitroprusside is a very potent substance which is used under clinically severe conditions (e.g. hypertensive crisis), while silymarin is used as a food supplement, which can have mild but clinically beneficial effects on blood pressure. The effects of silymarin are logically lower than the ones of a dedicated antihypertensive drug. EC50 of sodium nitroprusside is only 0.12 μM while that of our most potent substance, silychristin-19-O-sulfate is 19.2 μM. Anyway, we added this information to the Experimental part, to Figure 2 caption as well as to the Discussion together with similar data on a calcium channel blocker nifedipine.
The following text was added to Ex vivo experiments on isolated rat aorta (2.4) (at L. 138):
This concentration produces 99% vasorelaxation of rat aorta, and was calculated using the vasorelaxant curve of sodium nitroprusside obtained in our pre-experiments.
The following text was added to the Figure 2 caption (L.217):
The percentage of relaxation was calculated using the standard vasorelaxant drug sodium nitroprusside, which produce 99% vasorelaxation at 10 µM.
The following text was added to the Discussion (at L. 310):
Although the effects observed are incomparable with clinically used medication (EC50 value of silybin A was 26.8 μM; that of a calcium channel blocker nifedipine is 4.47 nM [35]), these results deserve attention since we are speaking about effects of a natural product which is used as a food-supplement and not about an approved drug. In particular…
Reviewer 2 Report
In this study, Jana Pourová and colleagues investigate the effects of flavolignans on platelet aggregation and blood vessels. Several in vitro studies were performed, and the results are well presented. In my opinion, the manuscript is of a overall good quality and it is easy and nice to read. I report below some comments/suggestions.
The abstract is very clear but in my personal opinion the number of keywords is very high, and some of them are rather general. Introduction is well written, even if maybe it appears to be slightly long in the first part (until line 61). I would suggest to move the figure showing the chemical structures of the compounds from the supplementary file to the manuscript. The author should also provide representative analytical data. Discussion, 4.1. Ex vivo experiments on isolated rat aorta: this role of natural flavonoids was previously investigated by other authors using similar experimental protocol. Authors could reference “Ribaudo G, Pagano MA, Pavan V, Redaelli M, Zorzan M, Pezzani R, Mucignat-Caretta C, Vendrame T, Bova S, Zagotto G. Semi-synthetic derivatives of natural isoflavones from Maclura pomifera as a novel class of PDE-5A inhibitors. Fitoterapia. 2015”. A brief discussion of the expected mechanism of action may also be provided. The concluding sentence in the discussion section “The main weakness of these results is the absence of a standard drug, which makes their comparison with other studies difficult” is not completely clear: why did de authors not introduce a standard compound in the study?
Author Response
The abstract is very clear but in my personal opinion the number of keywords is very high, and some of them are rather general.
Response
According to the instructions for authors of Nutrients, three to ten keywords are allowed. However, too general keywords and those overlapping the title were removed and partially replaced by better ones.
Introduction is well written, even if maybe it appears to be slightly long in the first part (until line 61).
Response
We re-considered this part of the Introduction to shorten it. It is not an easy task since we suppose that these data summarize the current status of research in this area and as a courtesy to the readers we prefer to keep it as such. If the reviewer insists, we can abbreiviate this section, however we would welcome an indication, which part(s) are redundant.
I would suggest to move the figure showing the chemical structures of the compounds from the supplementary file to the manuscript. The author should also provide representative analytical data.
Response
Figure S1 showing the chemical structures of the compounds was moved from the supplement to the main text (and all other figures were renumbered). Instead, the representative HPLC chromatograms of all compounds tested are now provided in the supplementary Figure S1.
Discussion, 4.1. Ex vivo experiments on isolated rat aorta: this role of natural flavonoids was previously investigated by other authors using similar experimental protocol. Authors could reference “Ribaudo G, Pagano MA, Pavan V, Redaelli M, Zorzan M, Pezzani R, Mucignat-Caretta C, Vendrame T, Bova S, Zagotto G. Semi-synthetic derivatives of natural isoflavones from Maclura pomifera as a novel class of PDE-5A inhibitors. Fitoterapia. 2015”.
Response
Thank you for suggesting this interesting paper whose reference was supplemented to the paper as ref. [34]. In fact, inhibition of PDE-5A represents one of possible mechanisms of action of the substances tested.
The following text was added to the Discussion (at 349):
Less expected mechanisms are also possible such as inhibition of phosphodiesterases that has been previously reported for some natural flavonoids [34].
A brief discussion of the expected mechanism of action may also be provided.
Response
We agree that the mechanism of action represents a crucial pharmacological aspect. We have shown that the vasorelaxation induced by silybin A is clearly dependent on the presence of intact endothelium. We plan to elucidate the mechanism of this vasorelaxation in details (e.g. dependence on NO generation, M receptor activation, role of potassium and/or calcium channels using standard antagonists such as L-NAME, atropine, TRAM-34, UCL-1684 or activators as BayK8644). These experiments however require a substantial time span and animals, so waiting for finishing of this analysis would significantly delay the revision (we are supposed to revise this manuscript within 5 days). For this reason, we plan to perform this complex analysis in our next article.
The following text was added to the Discussion (at L. 347):
Vasorelaxation can occur by various mechanisms, both endothelium dependent and endothelium independent. We have shown that the vasorelaxation induced by silybin A was clearly dependent on the presence of intact endothelium. Further experiments are necessary to elucidate this effect in details (e.g. NO generation, role of M receptors and/or of various potassium or L-type calcium channels). Less expected mechanisms such as inhibition of phosphodiesterases previously reported for some natural flavonoids are also possible.
The concluding sentence in the discussion section “The main weakness of these results is the absence of a standard drug, which makes their comparison with other studies difficult” is not completely clear: why did de authors not introduce a standard compound in the study?
Response
We agree that the concluding sentence was confusing. We meant that standard substances were not included in previously published studies. The presence of a standard substance is, of course, very important. Especially in the case of collagen-induced aggregation there are large differences in the results of the standard substance (ASA) between studies with EC50 ranging from 50 μM to 10 mM (Pyo, Planta Med. 2003; 69(3): 267-269; Oka Agric. Biol. Chem. 1986; 50(11): 2723-2727). We used standard compounds for comparison with tested flavonolignans in case of all antiplatelet measurements. In addition, we use double calibration: The dose of the inducer is first set to the minimal concentration, which causes maximal aggregation, and the second calibration is carried out using ASA and kaempferol to fit our standard curves. Thanks to this calibration it is also possible to compare the results obtained from measurements with blood from different donors. The sentence was modified to make it clear.
This change was made in the Discussion (at 446):
Sentence “The main weakness of these results is the absence of a standard drug, which makes their comparison with other studies difficult” was changed to “The main weakness of the studies published earlier is the absence of a standard drug, which makes their comparison with our results difficult”.
Other changes
Introduction (at 40):
Thoeprastus changed to Theophrastus.